# Optimization Workflow of Fumonisin Esterase Production for Biocatalytic Degradation of Fumonisin B_1_

**DOI:** 10.3390/life13091885

**Published:** 2023-09-08

**Authors:** Dániel János Incze, László Poppe, Zsófia Bata

**Affiliations:** 1Department of Organic Chemistry and Technology, Budapest University of Technology and Economics, Műegyetem Rakpart 3, H-1111 Budapest, Hungary; inczedani96@edu.bme.hu; 2Research and Development Laboratory, Dr. Bata Ltd., Bajcsy-Zsilinszky utca 139, H-2364 Ócsa, Hungary; 3Biocatalysis and Biotransformation Research Center, Faculty of Chemistry and Chemical Engineering, Babeş-Bolyai University of Cluj-Napoca, Strada Arany János 11, RO-400028 Cluj-Napoca, Romania

**Keywords:** enzyme, recombinant protein production, *Pichia pastoris*, media optimization, fumonisin esterase

## Abstract

Industrial enzyme production with the *Pichia pastoris* expression system requires a well-characterized production strain and a competitively priced fermentation medium to meet the expectations of the industry. The present work shows a workflow that allows the rapid and reliable screening of transformants of single copy insertion of the target production cassette. A constitutive expression system with the glyceraldehyde-3-phosphate dehydrogenase promoter (*pGAP*) with homology arms for the glycerol kinase 1 (*GUT1*) was constructed for the targeted integration of the expression plasmid in a *KU70* deficient *Pichia pastoris* and the production of a bacterial fumonisin esterase enzyme (CFE). A robust colony qPCR method was developed for the copy number estimation of the expression cassette. Optimization of the protein production medium and the scale-up ability was aided by design of experiments (DOE) approach resulting in optimized production conditions at a semi-industrial scale. A novel fermentation medium containing 3% inactivated yeast and 2% dextrose in an ammonium-citrate buffer (IYD) was shown to be a promising alternative to YPD media (containing yeast extract, peptone, and dextrose), as similar protein titers could be obtained, while the cost of the medium was reduced 20-fold. In a demonstration-scale 48 h long fed-batch fermentation, the IYD media outperformed the small-scale YPD cultivation by 471.5 ± 22.6%.

## 1. Introduction

Enzymes can catalyze various kinds of industrially relevant reactions, typically under mild conditions, with high selectivity and efficiency, and with nontoxic ingredients [1]. However, volumetric productivity and production cost are usually among the factors that hinder the use of enzymes as biocatalysts as an industrially relevant alternative to chemical methods [1]. In recent decades, the methylotrophic yeast *Pichia pastoris* (*P. pastoris*) became one of the most important host strains for recombinant protein production because it can reach a high cell density [2] while growing in a relatively inexpensive media [3,4] and due to its ability to secrete the proteins into the fermentation media at high titers.

In most cases, the alcohol oxidase 1 promoter (*pAOX1*) is used for the construction of expression vectors in *P. pastoris*, as it is a tightly regulated, strong promoter that can be induced by methanol (MeOH) [2,3,4,5]. However, there are several drawbacks to using pAOX1 in large-scale production, such as the toxicity, fire hazard, and cost of MeOH, as well as the need to switch from feeding glycerol/dextrose to MeOH during fermentation [4]. To overcome these obstacles, several other promoters have been studied. The glyceraldehyde-3-phosphate dehydrogenase promoter (*pGAP*) is the most well-characterized, growth-coupled, constitutive promoter for heterologous protein production in *P. pastoris* [4,6,7]. *pGAP* becomes a favorable promoter in the *P. pastoris* expression system for large-scale production due to its lack of the mentioned negative aspects of the *pAOX1* promoter. Moreover, in several cases, *pGAP* even outperforms *pAOX1* [4,7,8,9].

The integration site of the linearized production plasmid can strongly influence the integration and expression efficiency [10]. Targeted integration of the production plasmid can eliminate the undesired and unknown effects of random integration in other important loci of the genome [11]. The proper integration of exogenous DNA with 50 bp homology arms is almost 100% efficient in *Saccharomyces cerevisiae*, as the preferred double-stranded break (DSB) repair mechanism in baking yeast is the homologous recombination (HR). Contrarily, in *P. pastoris*, non-homologous end joining (NHEJ) is the preferred mechanism which leads to unspecific integration and possibly insertions and deletions [12], and HR occurs only at frequencies of 0.1 to 30%, even with 1000 bp homology arms [11]. The frequency of HR in the DSB repair mechanism in *P. pastoris* can be enhanced by the deletion of the *ku70* gene [11], to reach more than 90% targeting efficiencies with 250–650 bp homology arms. Consequently, *KU70*-deficient (*ΔKU70*) *P. pastoris* strains support the efficient targeting of production plasmids [11,13].

The investigation of the copy number of the expression cassettes in *P. pastoris* with quantitative polymerase chain reaction (qPCR) has been reported [14]; the SYBR Green method using the *ARG4* as a reference gene yielded appropriate results [14]. However, that method required the isolation of the genomic DNA (gDNA) of the screened transformants. Colony PCR is a frequently used method for the rapid screening of colonies, without the need for gDNA isolation. Successful colony PCR methods have been reported for yeast cells, such as *Saccharomyces cerevisiae*, *Kluyveromyces lactis* [15], and *Pichia pastoris* [16]. These methods included standard PCR reactions using Taq DNA Polymerase Master Mix (abm, Canada) [16] or Phire™ Plant Direct PCR Master Mix (Thermo Scientific, Waltham, MA, USA) [15], and the addition of the cells directly into the reaction mixture. However, debris from yeast cells and some media components can inhibit the PCR polymerase [15].

Selecting the optimal culture medium is one of the key elements of the high-level expression of recombinant proteins in *P. pastoris*. The fermentation medium for *P. pastoris* can be either complex or defined. Complex media are usually nutrient-rich and contain micronutrients, vitamins, and metabolic intermediates [17]. Typically, complex media consist of dextrose or glycerol, yeast extract, and peptone (like YPD or BMGY media [18]), although industrial byproducts, such as corn steep liquor powder, fish meal, soybean meal, peanut cake powder [19], extruded bean [20], and maize meal or bean pulp [21] have been successfully used as a medium component as well. Defined media are composed of pure chemical substances, for instance, simple organic carbon sources (such as dextrose and glycerol), inorganic salts, vitamins, amino acids, and ammonia. The most used defined media for *P. pastoris* fermentation is the basal salts medium (BSM) [22], although this medium was modified several times to increase per-cell productivity [17,23,24]. The largest advantage of defined media is its batch-to-batch consistency; furthermore, defined media are usually cheaper than complex media, especially those containing laboratory-grade yeast extract and peptone [17]. However, during fermentation in defined media, the yeast must synthesize all the metabolic intermediates, mostly resulting in slower growth rates and lower product yields than in complex media [17]. Thus, when the application of the recombinant protein does not require complicated downstream processes and high purity, complex media containing low-cost industrial byproducts are often advisable [19].

The present work shows a workflow that allows the rapid and reliable screening of transformants of single copy insertion of the target production cassette, followed by a design of experiments (DOE) approach for media optimization of the protein production and the scale-up ability of the optimized production conditions to a semi-industrial scale. The workflow is applied to the fumonisin esterase as a biocatalyst for fumonisin B_1_ degradation.

Fumonisins are a group of prevalent mycotoxins produced by several species of *Fusarium* molds, such as *Fusarium verticilloides* and *Fusarium proliferatum*, and these mycotoxins are found predominantly in corn, maize, and corn-based animal feeds worldwide [25]. Fumonisin B_1_ (FB_1_) is the primary and most toxic member of the fumonisin family [26,27]. FB_1_ is chemically characterized as an aliphatic hydrocarbon with a terminal amine group and tricarboxylic acid (TCA) ester side chains (Figure 1).

The economic impact of animal feed is enormous because previous research estimated the annual financial losses in the USA due to FB_1_ in animal feed to be USD 30–46 million in an outbreak year of Fusarium ear rot [28]. FB_1_ has a similar structure to that of sphingosine and sphinganine, making the aminopentol backbone compete against the binding of the sphingoid base substrate. In contrast, while the TCA side chain interferes with the binding of the fatty acyl-CoA, the accumulation of free sphingoid bases via the inhibition of ceramide synthase leads to the disruption of the sphingolipid metabolism [29,30]. Fumonisin B_1_ causes various toxic effects in livestock and poultry, including poor growth, neurotoxicity, immunotoxicity, reproductive toxicity, and tissue and organ toxicity [31,32].

Mycotoxins have high heat stability; thus, they are challenging to remove from the food or feed chain and can contaminate the entire manufacturing process [33]. Biocatalytic degradation employs enzymes to break the toxic chemical structures of mycotoxins to produce products of lower or no toxicity [34]. Fumonisin esterases (FE) catalyze the consecutive de-esterification of FB_1_ at the C-6 and C-7 positions (Figure 1), resulting in an aminopentol (hydrolyzed FB_1_, HFB_1_) and two tricarballylic acid (TCA) molecules as final products [35].

## 2. Materials and Methods

### 2.1. Strains, Vectors, and Media

*Pichia pastoris* BSY10dKU70 strain and pBSY1S1G plasmid were purchased from Bisy GmbH (Hofstätten an der Raab, Austria). The nucleic acid sequence of fumonisin esterase from *Caulobacter* sp. ATCC 55552 (further referred to as CFE) [36] was codon-optimized (Table A1) based on *P. pastoris* codon bias and purchased from ATUM (Newark, CA, USA), cloned into pD912 vector. *Escherichia coli* DH5α was used as the bacterial host for the plasmid vector amplification, and the *P. pastoris* BSY10dKU70 strain was selected as the expression host. *P. pastoris* cells were grown in YPD medium (1% yeast extract, 2% peptone, and 2% dextrose), and the LB medium (1% tryptone, 0.5% yeast extract, 0.5% NaCl) was used for *E. coli* DH5α cultivation. The media was supplemented with Zeocin™ (InvivoGen, San Diego, CA, USA). In the case of *E. coli* and *P. pastoris* selection, Zeocin concentrations were fixed at 50 μg mL^−1^, and 100 μg mL^−1^, respectively. Methanol (MeOH), water (H_2_O) (all LC-grade), as well as concentrated hydrochloric acid solution (cc. HCl) and all reagents for culture media and derivatization, buffer salts, and bovine serum albumin (BSA) were purchased from either Reanal Laborvegyszer Kft. (Budapest, Hungary) or Merck KGaA (Darmstadt, Germany).

### 2.2. Construction of the Production Plasmid

The production plasmid pGAPGUT1-CFE was designed for the highly efficient targeted integration of the plasmid at the glycerol kinase 1 (*GUT1*) locus of the *P. pastoris* BSY10dKU70 strain and the constitutive expression of CFE with the glyceraldehyde-3-phosphate dehydrogenase promoter (*pGAP*). The improved targeting efficiency via the increased homologous recombination frequency can be associated with the deleted *KU70* gene in this particular strain [11]. The plasmid consists of a production cassette with a *pGAP* constitutive promoter, the cDNA of CFE, and an *AOX1* termination sequence; a Zeocin selection cassette; a bacterial replication origin; and 5′ and 3′ homologous integration sequences of the glycerol kinase (*GUT1*) gene of *P. pastoris* (Figure A1). The production cassette, the selection cassette with the bacterial replication origin, and the homologous integration sequences were PCR-amplified from the pBSY1S1G plasmid, the pD912 vector, and the gDNA of *P. pastoris* BSY10dKU70 strain, respectively. The PCR reactions were performed with a Phusion High Fidelity PCR Master Mix with an HF buffer (New England Biolabs, Ipswich, MA, USA, further referred to as NEB). The primers used in this work are listed in Table A2. The PCR products were gel-purified with QIAquick Gel Extraction Kit (Qiagen, Hilden, Germany), and pGAPGUT1 plasmid was assembled with Gibson Assembly Master Mix (NEB), according to the manufacturer’s protocol. The assembly product was directly transformed into chemically competent *E. coli* DHL5α cells—prepared in-house according to the Hanahan method [37]. The transformed *E. coli* cells were selected on LB agar (0.5% *w*/*v* yeast extract, 1% *w*/*v* tryptone, and 1% *w*/*v* sodium chloride, 1% *w*/*v* bacteriological agar) supplemented with 50 μg mL^−1^ Zeocin. LB medium with 50 μg mL^−1^ Zeocin was used for plasmid amplification. The plasmids were isolated with the EZ-10 Spin Column Plasmid DNA Kit (Bio Basic Canada Inc., Markham, ON, Canada), according to the manufacturer’s protocol, and subsequently sequenced to confirm that the assembly was successful. The cDNA of CFE was PCR-amplified from the pD912-CFE vector. This PCR product and pGAPGUT1 were digested with SapI restriction endonuclease (NEB) and ligated with T4 DNA ligase (Thermo Scientific, Waltham, MA, USA) to produce pGAPGUT1-CFE. The transformation, selection, plasmid amplification, and purification were performed as previously described.

### 2.3. Preparation of the CFE-producing P. pastoris Strain

For the heterologous production of CFE, pGAPGUT1-CFE was linearized with NruI restriction endonuclease (NEB). The production of electrocompetent *P. pastoris* cells, transformation, selection, and protein production was performed according to ATUM’s protocol [18]. Only 500 ng of the linearized pGAPGUT1-CFE plasmid was transformed into 100 µL of electrocompetent *P. pastoris* BSY10dKU70 suspension, to promote single-copy integration [11]. Cells with successfully integrated plasmid were selected on YPDS agar plates (1% yeast extract, 2% peptone, 2% dextrose, 1M sorbitol, 1% bacteriological agar) supplemented with 100 μg mL^−1^ Zeocin. Four colonies—chosen for further screening—were grown on BMGY agar (1% yeast extract, 2% peptone, 100 mM potassium phosphate buffer pH 6.0, 1% glycerol, 0.134% Yeast Nitrogen Base with amino acids, 1% bacteriological agar) to evaluate the correct integration via the consequent *GUT1* gene knockout and disabled growth on glycerol.

### 2.4. Evaluation of Transgene Copy Number and Integration Site with qPCR

For the qPCR measurements, the *ARG4* gene was chosen as an endogenous control gene, and *GUT1* as the target gene. Therefore, primer pairs—being specific to *GUT1* and *ARG4* genes and producing PCR products of 88 bp and 84 bp, respectively—were generated (Table A2). For the qPCR measurements, gDNA was isolated with a Quick-DNA Fungal/Bacterial Miniprep Kit (Zymo Research, Irvine, CA, USA) from an overnight culture of a *wt* and of a pGAPGUT1-CFE-transformed *P. pastoris* colony. The qPCR measurements were performed in CFX96 Touch Real-Time PCR Detection System (Bio-Rad Laboratories, Hercules, CA, USA), using SYBR™ Select Master Mix for CFX (Applied Biosystems, Waltham, MA, USA), in reaction volumes of 10 µL, at primer concentrations of 250 nM, and 1 ng of gDNA. 10, 1, 0.1, and 0.01 ng of *wt* gDNA were used for the generation of the calibration curves for both *ARG4* and *GUT1*, which are equal to copy numbers 9.82 × 10^5^, 9.82 × 10^4^, 9.82 × 10^3^, and 9.82 × 10^2^, respectively, assuming the genome size of *P. pastoris* is 9.43 Mbp [38]. The measurements were performed in triplicate. The C_T_ values and the copy numbers of pGAPGUT1-CFE-transformed *P. pastoris* were calculated by the software of the qPCR system.

### 2.5. Rapid Colony qPCR Analysis of the Copy Numbers

Colony qPCR using the Pfaffl method [39] for relative quantification could be a less reagent- and time-consuming analysis of the copy numbers of the integrated plasmid. This method allows one to determine the relative copy number of a target gene in a test sample to a known calibrator sample using a reference gene as a normalizer, without knowing the exact copy numbers or DNA quantities [39]. The single-copy colony with targeted integration evaluated in Section 2.4 (further referred to as colony Nr. 1) was used as a calibrator sample for the analysis. Three single colonies of pGAPGUT1-CFE-transformed *P. pastoris* from the selection plate were collected with sterile disposable inoculating loops and resuspended in 30 µL of sterile distilled water. The cell suspensions were incubated at 95 °C for 10 min, vortexed, and either centrifugated at 3000 rcf for 1 min or left the cells in suspension. For more accurate results, qPCR primers specific to the Zeocin resistance gene were used as a target with a PCR product of 86 bp, alongside the primers specific to the *ARG4* and *GUT1* genes. The qPCR analyses were performed as described in Section 2.4, with the modification that 1 µL of the supernatants or 1 µL of the suspensions of the boiled cells was added to the reaction mixture as the DNA samples. The amplification efficiencies of the target and reference genes are usually different, which could lead to wrong results. To calculate the efficiencies of the reactions, a 10-fold dilution series (1×, 10×, 100×, 1000×) was created from the supernatant of colony Nr.1. The qPCR measurements were performed in triplicates. The C_T_ values were calculated by the software of the qPCR system and were plotted against the log of the copy numbers. The copy numbers were unknown, but due to the 10-fold dilution of the calibration sample and the fact that only the slope of the fitted linear equation is used for the calculation of the efficiency, the logs of the copy numbers were assumed as 4, 3, 2, and 1. Efficiency (E) was calculated as follows:E = 10^−(1/slope)^,(1)
where E is the amplification efficiency of the qPCR reaction, and slope is the slope of the linear equation fitted to the calibration C_T_-log copy number data points [39]. To determine the copy number ratio (Ratio) between the test and calibrator samples, the following equation can be used:Ratio = (E_target_^(ΔCT, target (calibrator-test))^)/(E_ref_^(ΔCT, ref (calibrator-test))^),(2)
where E_target_ and E_ref_ are the amplification efficiencies of the target and reference genes, respectively. ΔC_T_, target (calibrator-test) is the C_T_ of the target gene in the calibrator minus the C_T_ of the target gene in the test sample, and ΔC_T_, ref (calibrator-test) is the C_T_ of the reference gene in the calibrator minus the C_T_ of the reference gene in the test sample [39]. The standard ΔΔC_T_ method without correction assumes all E values to be 2; therefore, with this method, the ratio can also be calculated with Equation (2) by simply replacing E_target_ and E_ref_ with 2 [39].

### 2.6. Optimization of a Novel Fermentation Medium

A novel fermentation medium was designed to test the potential of feed-grade inactivated yeast as a media component of industrial *P. pastoris* fermentation and recombinant CFE production. A three-level two-factor full factorial design with 3 repetitions at the central point was applied to study the relationship between CFE production and two potential factors, namely the initial concentration of inactivated yeast (*w*/*v*%, further referred to as IY%) and initial concentration of dextrose (*w*/*v*%, further referred to as S%). The design proposal is detailed in Section 3.3.

All fermentation media contained 50 mM citric acid, 300 mM NH_4_OH, as well as feed-grade inactivated yeast and dextrose at various concentrations. The pH of the media was adjusted to 6.0 with cc. HCl solution. Media without dextrose and 50 *w*/*w*% dextrose solution were autoclaved separately for 15 min at 121 °C, and the proper amount of dextrose solution was added to the dextrose-free medium after the media had cooled down. A reference fermentation in a modified YPD medium (1% yeast extract, 2% peptone, 4% dextrose) was performed in triplicate. For the optimization, 20 mL volumes of fermentation media were applied in 100 mL baffled flasks, which were inoculated with 2 mL volumes from an overnight culture of a *P. pastoris* colony containing an integrated single copy of pGAPGUT1-CFE production plasmid and shaken in a shaking incubator at 200 rpm at 28 °C for 24 h. After 24 h, 1 mL samples were taken from the fermentations, the samples were centrifugated at 13,000 rpm for 1 min, and the supernatants were analyzed. The results of the full factorial design experiment were analyzed with Statistica (v.14, StatSoft, Hamburg, Germany). A significance level of *p* < 0.05 was considered significant. The *t*-test was used to evaluate the significance of the regression coefficients. Fitted surface diagram and data plots were generated by the Statistica software.

Fermentations were conducted in triplicate at the calculated optimal inactivated yeast % and dextrose % levels to validate the model.

### 2.7. Test Fermentation in a 5 L Fermenter

The scalability of the optimized fermentation media (50 mM citric acid, 300 mM NH_4_OH, 6 *w*/*v*% dextrose, 4 *w*/*v*% inactivated yeast, pH adjusted to 6.0 with cc. HCl, in tap water, 0.2 mL of Componenta FG anti-foaming agent (Ecolab, Saint Paul, MN, USA)) was tested on a 5 L scale. Thus, 4.5 L of dextrose-free fermentation media was autoclaved for 1 h at 121 °C in an Univessel Glass 5 L double wall fermenter (Sartorius, Göttingen, Germany). After the media had cooled down, 0.6 kg of 50 *w*/*w*% dextrose solution (autoclaved separately for 15 min at 121 °C) was added to reach 5 L of final volume. The fermenter was inoculated with 500 mL of an overnight culture of *P. pastoris* containing an integrated single copy of pGAPGUT1-CFE production plasmid in YPD media. The fermentation was implemented at 28 °C, at 300 rpm agitation speed, at 0.5 L air min^−1^ L media^−1^ aeration, and pH was kept at 6.0 with 25 *v*/*v*% NH_4_OH solution, controlled by a Biostat B-DCU Bioprocess System (Sartorius, Göttingen, Germany). After 24 h, a 10 mL sample was taken, centrifugated at 13,000 rpm for 1 min, and analyzed for FE activity.

### 2.8. Scaling-Up of the Fermentation to 100 L

The fed-batch cultivation in a 160 L fermenter (Április 4. Gépipari Művek, Kiskunfélegyháza, Hungary) was implemented with 100 L of initial medium containing 50 mM citric acid, 300 mM NH_4_OH, 2 *w*/*v*% dextrose, 3 *w*/*v*% inactivated yeast, in tap water, and 500 mL Componenta FG. For this fermentation, the pH of the medium was adjusted to 6.0 with cc. HCl and the fermentor containing 96 L dextrose-free medium were sterilized for 45 min at 120 °C. After the medium has cooled down to 28 °C, 4 kg of 50 *w*/*w*% dextrose solution (autoclaved separately for 15 min at 121 °C) was added. The fermenter was inoculated with 5 L of a 24 h culture of *P. pastoris* containing an integrated single copy of pGAPGUT1-CFE production plasmid in YPD media. The fermentation was carried out at 28 °C, at 150 rpm agitation speed, at 1 L air min^−1^ L media^−1^ aeration, 0.8 bar tank pressure, and pH was kept at 6.0 during the whole fermentation with the addition of 25 *v*/*v*% NH_4_OH solution. After 24 h, 3 × 10 mL samples were taken from the fermenter. Subsequently, 4 kg of 50 *w*/*w*% dextrose solution was added to reach 2 *w*/*v*% dextrose concentration in the fermenter, and 50 *w*/*w*% dextrose solution was started to be fed for an additional 24 h at 720 g h^−1^ constant feeding rate. After a total of 48 h fermentation, 3 × 10 mL samples were taken and analyzed for FE activity.

### 2.9. Analysis of the Fermentations

#### 2.9.1. SDS-PAGE

Fermentation supernatants were analyzed by sodium dodecyl sulfate-polyacrylamide gel electrophoresis (SDS-PAGE), using 4–15% Mini-PROTEAN^®^ TGX™ Precast Protein Gels (Bio-Rad, Hercules, CA, USA). SDS PAGE analysis of a representative set of CFE fermentations is shown in Figure A2.

#### 2.9.2. Enzyme Activity Measurement

All substrate and enzyme solutions were diluted for the enzyme activity measurements in buffer A (50 mM potassium-phosphate buffer supplemented with 100 µg mL^−1^ BSA, at pH = 6.0). Analytical standard FB_1_ solution was diluted to a final concentration of 100 mg L^−1^. The enzymatic reaction was carried out in a plastic tube of 200 µL, using 72.2 µL of the FB_1_ solution, 7.8 µL of buffer A, and 20 µL of enzyme solution. Before the reaction, the crude fermentation supernatant was diluted to a concentration that resulted in a conversion of the enzyme reaction to less than 30% within 15 min. Table A3 shows the dilution factors of each fermentation analyzed in this work. The reaction mixtures were incubated at 37 °C in a TDB-120 Dry Block Thermostat (Biosan, Riga, Latvia) for 15 min and the reactions were terminated addition of 180 µL of MeOH to 20 µL portion of the reaction mixture, and analyzed with HPLC-FLD.

#### 2.9.3. HPLC-FLD Analysis

A sample solution of 10 µL was mixed with 10 µL of OPA reagent (40 mg of *o*-phthalaldehyde and 50 µL of β-mercaptoethanol in 1 mL of MeOH and 5 mL of 0.1 M Na_2_B_4_O_7_ solution) and kept at 20 °C for 5 min before injection. The derivatized analytes were separated on a reverse phase C18 column (Kinetex, 2.6 µm, C18, 100 Å, 100 × 4.6 mm) (Phenomenex, Torrence, CA, USA) in a Shimadzu CTO-40C column thermostat (Shimadzu Corporation, Kyoto, Japan) at 30 °C, and analyzed with a Shimadzu HPLC device coupled to a fluorescent detector (Shimadzu RF-20A XS, extension wavelength= 335 nm, emission wavelength = 440 nm). The mobile phase was fed via a Shimadzu LC-40D XR pump at a flow rate of 0.75 mL min^−1^. Initially, pure Eluent A (MeOH: 50 mM sodium phosphate buffer (pH 5.0), 65:35) was pumped for 5 min, followed by a linear gradient of Eluent B up to 70% (HPLC-grade methanol) within 9 min. Finally, the column was re-equilibrated for 3 min with 100% of Eluent A. The concentration of FB_1_ and HFB_1_ was determined from standard calibrations. FB_1_ and HFB_1_ showed the same molar response factor, as the signal originated from the OPA derivate, which was formed with the molecules’ amino group and seemingly independent of the presence of the TCA groups. Therefore, the same molar response factors were applied for the calculation of pHFB_1_ concentrations. A representative HPLC trace of the analysis of the enzymatic hydrolysis of FB1 is depicted in Figure A3.

#### 2.9.4. Calculation of the Relative Activity %

The relative activity % was calculated by the equations below:Relative activity % = (((X_FB1,sample_ × D_sample_)/((X_FB1,control_) × D_control_)) × 100%,(3)
X_FB1_ = c_FB1_/(c_FB1_ + c_pHFB1_ + c_HFB1_),(4)
where D_sample_ and D_contol_ are the dilutions of the sample and the control in the enzyme reaction, respectively, c_FB1_,c_pHFB1,_ and c_HFB1_ are the concentrations of FB_1_, pHFB_1_, and HFB_1_, respectively.

## 3. Results and Discussion

### 3.1. Preparation of a P. pastoris Strain Containing an Integrated Single-Copy of pGAPGUT1-CFE Production Plasmid

One of the aims of our study was to create a production plasmid that can serve as an *E. coli-P. pastoris* shuttle vector with an antibiotic selection marker which allows the strong constitutive secretion of a model protein (CFE) and involves 3′ and 5′ flanking sequences of a *P. pastoris* gene for the enhanced targeted integration of the plasmid. For the targeting locus, *GUT1* was chosen, as the knockout of the *GUT1* gene leads to abolished growth on glycerol but does not influence the utilization of dextrose [10,11], which facilitates the easy selection of colonies with correctly targeted integration. The pGAPGUT1-CFE production plasmid was successfully assembled with the Gibson Assembly of five PCR products, and seamless SapI cloning of the cDNA of CFE (Figure A1).

The 6468 bp linearized pGAPGUT1-CFE was successfully used to transform the *P. pastoris* BSY10dKU70 strain. All four selected transformants showed disabled growth on BMGY agar, which indicated the disruption of the *GUT1* gene, and the correct integration of the pGAPGUT1-CFE plasmid.

### 3.2. qPCR Analysis of the Transformants

#### 3.2.1. Absolute Quantification

Both in the genomes of the wild-type (*wt*) and pGAPGUT1-CFE single-copy-integrated *P. pastoris*, the proportion of the *GUT1*:*ARG4* genes would be 1:1. Should multiple copy integration occur, the *GUT1*:*ARG4* proportion would be X:1 (X being the number of integrated copies). The isolated gDNA of colony Nr. 1 was analyzed with qPCR using SYBR Select Master Mix, and *wt P. pastoris* gDNA was used for the calibration. The calculated *GUT1*:*ARG4* proportion was 1.22 ± 0.03:1 (Table 1), which confirms the single-copy integration.

#### 3.2.2. Relative Quantification Using the Pfaffl Method

A characterized pGAPGUT1-CFE single-copy-integrated *P. pastoris* (colony Nr. 1) can serve as a calibrator sample for the relative quantification using the Pfaffl method. The three other colonies, all showing disabled growth on glycerol as the sole carbon source were analyzed as samples. The *GUT1* and *ZEO* genes were the target genes, whereas *ARG4* was the reference gene since analysis results of two reference genes instead of one reduces the uncertainty of the results. This combined method utilizes the advantages of the colony PCR and the Pfaffl method [39], namely the unnecessary gDNA isolation and purification and the correction with the amplification efficiencies of the different gene targets, respectively.

Interestingly, when the boiled colonies were added to the reaction in suspension, no PCR amplification could be detected (Figure 2). However, when only the supernatant of the boiled colony suspension after centrifugation was added, the PCR amplification was successful. This can be explained by inhibitory interactions of cell debris with the polymerase in the PCR [15]. Thus, it is advisable to centrifugate the boiled cell suspensions and add only the supernatant to the qPCR reaction mix.

Accordingly, the supernatants of the boiled colonies were added to the colony qPCR reactions in the following experiments. To calculate the efficiencies of the reactions, a 10-fold dilution series (1×, 10×, 100×, 1000×) was created from the supernatant of colony Nr. 1. The amplification efficiencies (E) of *ZEO*, *GUT1,* and *ARG4* genes were calculated with Equation (1), using the slopes of the fitted linear of the calibration data (Figure 3).

E values of 2.2727, 2.0671, and 2.2831 were found for the *ZEO*, *GUT1*, and *ARG4* genes, respectively. The relative copy numbers (*ZEO:ARG4* and *GUT1:ARG4*) were calculated with Equation (2), both by using the calculated E values (Pfaffl method) or estimating E to be 2 (ΔΔC_T_ method without correction) (Table 2). The relative copy numbers, ranging from 0.7 to 1.4, indicated that all four colonies integrated a single copy of the plasmid. These results are to the expectations of having only one copy at the correct locus due to the low DNA quantity used for the transformation and the high targeting efficiency of the *P. pastoris* BSY10dKU70 strain. Such degrees of deviation from 1 are within the uncertainty of the method. The Pfaffl method did not result in a more accurate calculation; however, it is still advisable to consider the E values, as more significant differences among them can cause significant inaccuracies, especially at high copy numbers [39].

These results indicated that the presented colony qPCR using the relative quantification method can be suitable for the evaluation of the correct targeted integration of the pGAPGUT plasmid.

### 3.3. Optimization of a Novel Fermentation Media for Enzyme Production

Dozens of different enzymes for food or feed additives have been successfully produced by *P. pastoris* [5]; however, the production cost can be a burden on their industrial applicability. To significantly reduce enzyme production costs, a novel fermentation media utilizing inactivated yeast as a major component was designed for extracellular recombinant enzyme production by *P. pastoris*. As the target enzyme for evaluation, CFE, a mycotoxin-degrading biocatalyst, was chosen as the protein to be produced. Inactivated yeast has great potential as a fermentation media component and has been successfully used as such in lactic acid bacteria cultivation [40]. However, to the best of our knowledge, inactivated yeast has not been utilized as a media component for recombinant protein production in *P. pastoris* to date.

A three-level two-factor full factorial design was employed to optimize the fermentation for CFE production. The media contained ammonium citrate buffer with the same ammonium concentration (300 mM) as used in the media developed by D’Anjou and Daugulis [24], as well as dextrose and inactivated yeast at various levels. The FE activities of the fermentation supernatants were measured, and the activity of the fermentation in the YPD medium was defined as 100% for relative activity (RA%) determination. Inactivated yeast and dextrose content of the media (IY% and D%, respectively) were the independent factors, and the relative enzyme activity (RA%) was the dependent variable (Table 3).

The results indicated the strong influence of the different media on the CFE expression (Figure A2), as the RA% values varied between 17.0% and 101.1% (Table 3). A second-order model was fitted to the results with Statistica software. Linear and quadratic effects were computed, two-way interactions were allowed, and only significant effects were used for the model. According to the results, the linear and quadratic effects of the inactivated yeast % (IY% (L) and IY% (Q), respectively), the linear effect of the dextrose % (S% (L)) and the interaction between the Inactivated yeast % linear and dextrose % linear effects were significant (*p* < 0.05) (Table 4). The fitted quadratic model is adequate with an F-probe, as the *p*-value of the lack of fit is *p* = 0.256 > 0.05 (Table 4). The determination coefficient (R^2^ = 0.994) and the adjusted coefficient of determination (Adj R^2^ = 0.990) indicate that the model sufficiently describes the correlation between the independent and dependent variables.

The regression coefficients were also computed by the software (Table 5), and the model can be described with the following equation:Relative activity% = 16.22 + 36.48 × IY% − 4.87 × IY%^2^ + 1.03 × IY% × S%(5)

The fitted surface of the model shows that the maximal RA% can be reached at 6% dextrose and 4% inactivated yeast (Figure 4). Using Equation (5), an RA% of 121.1 ± 8.7% is the mean of experimental values, further validating the model and indicating its applicability. This corresponds to a 21.3 mg L^−1^ expression level, calculated by using the specific activity of CFE described in our parallel study [41]. Scaling up using the 6% dextrose and 4% inactivated yeast concentrations to 5 L scale fermentation resulted in RA% = 114.2%, further confirming the validity of the model during upscaling of the fermentation.

### 3.4. Scale-Up of the CFE Production in IYD Media

#### 3.4.1. Economic Calculation for Fermentation Scale-Up

Laboratory-scale fermentations usually aim for the highest production yield, regardless of the cost of the media, as the labor cost and other fixed costs significantly outweigh the cost of the production media. On the other hand, when scaling up, and in industrial processes, the cost of the media can be as high as 75% of the total production cost; thus, the cost factor must also be considered when the fermentation is scaled up. Therefore, the relative cost of the media in the above experiment is compared in Table 6.

From the calculation, it is clear that the laboratory media made up of laboratory-grade ingredients is at least 12.6 times more expensive than all the tested media (Table 6).

The solver function of Excel was used to find the optimal media composition maximizing the FE activity by the model of Equation (5) and minimizing the price of the media. The program was limited to 0–6% for IY% and 2–6% for D% to avoid extrapolation. The solver suggested a composition of 3.32 IY% and 2 D% for the Optimal choice. The model predicts an RA% of 90.5% for this media composition. Moreover, the optimal IYD medium is more than 20 times less expensive than YPD, with less than 10% lower activity (Table 6).

#### 3.4.2. Scale-Up to Demonstration Scale

In the 100 L demonstration scale, 2% dextrose and 3% inactivated yeast were used due to technical difficulties of adding precisely 3.32% IY. Interestingly, the obtained RA% in the batch fermentation after 24 h was 144.6 ± 4.7%, which is significantly higher than the predicted (90.5%). Better aeration due to the fermenter geometry, agitation speed, aeration volumetric speed, or the applied backpressure could result in a more efficient fermentation and higher titer. After the extra 24 h fed-batch fermentation, an RA% of 571.5 ± 22.6% was obtained. This expression level of CFE is comparable to the one obtained in our parallel study [41] using a methanol-induced pAOX1 expression system in an unoptimized one-week-long lab-scale fermentation (according to the protocol of ATUM [18]) in buffered complex glycerol medium (BMGY).

## 4. Conclusions

A constitutive expression system was successfully constructed for the targeted integration of the expression plasmid to the glycerol kinase 1 (*GUT1*) locus in a *KU70* deficient *Pichia pastoris* and the production of a bacterial fumonisin esterase enzyme (CFE). The rapid colony qPCR method for the copy number estimation of the integrated expression cassette presented in this study can facilitate the high throughput screening of the transformants and can be considered for other production strains as well. Feed-grade inactivated yeast showed great potential to replace the bacteriological-grade yeast extract and peptone in laboratory-scale and demonstration-scale fermentations. Given the excellent results, it is advisable for all laboratories to expand their reagent use to food- and feed-grade ingredients.

## Figures and Tables

**Figure 1 life-13-01885-f001:**
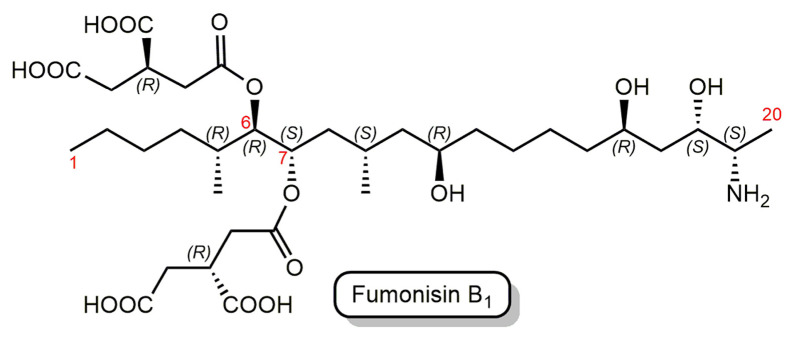
Structure of fumonisin B_1_ (FB_1_). The parent chain numbering is shown by red numbers.

**Figure 2 life-13-01885-f002:**
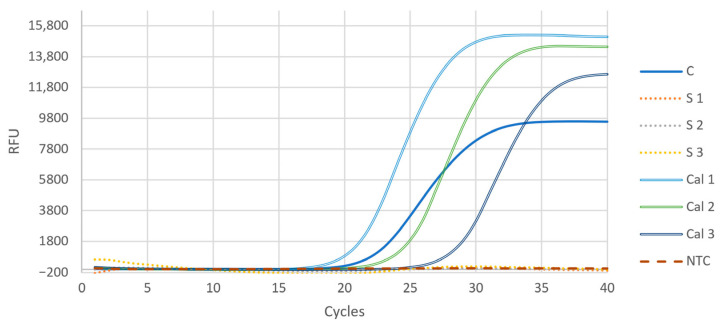
Comparison of the two methods for colony qPCR. When the supernatant of the boiled colony was added to the reaction mixture (C), PCR amplification was successful. When the cell suspension was added directly into the reaction, no amplification could be detected (S 1–3). Calibrations were performed with the isolated gDNA of colony Nr.1; copy numbers were 9.82 × 10^5^, 9.82 × 10^5^, and 9.82 × 10^5^ for Cal 1, Cal 2, and Cal 3, respectively. NTC is the no template control measurement. ZEO was the target gene in all reactions demonstrated in this figure. RFU = relative fluorescent unit.

**Figure 3 life-13-01885-f003:**
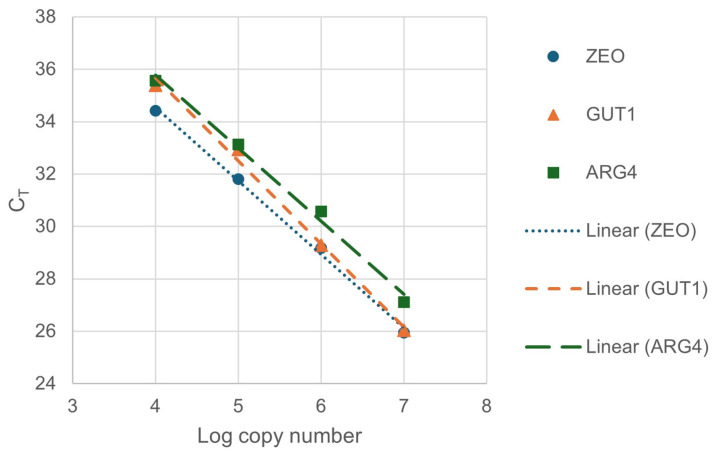
Calibration lines for calculation of amplification efficiencies. For the fitted lines of *ZEO*, *GUT1*, and *ARG4* genes, the slopes were −2.8047, −3.1709, and −2.7892; the R^2^ values of the fittings were 0.9972, 0.9945, and 0.9925, respectively.

**Figure 4 life-13-01885-f004:**
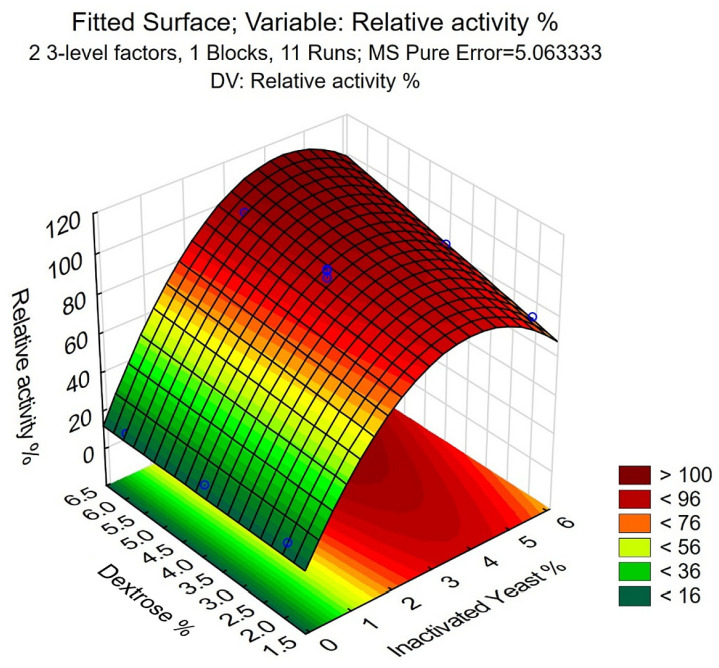
The fitted surface of the model for the optimization of inactivated yeast–dextrose media, generated with Statistica. The experimental values are indicated with blue circles.

**Table 1 life-13-01885-t001:** Results of the absolute quantification of colony Nr. 1, a pGAPGUT1-CFE single-copy-integrated *P. pastoris* colony.

Replicate	*ARG4* C_T_	*GUT1* C_T_	*ARG4* Copy Number	*GUT1* Copy Number	*GUT1:ARG4*
1	22.27	21.78	10.21 × 10^5^	12.82 × 10^5^	1.25
2	22.24	21.79	10.45 × 10^5^	12.78 × 10^5^	1.22
3	22.26	21.86	10.34 × 10^5^	12.20 × 10^5^	1.18

**Table 2 life-13-01885-t002:** Results of the relative quantification of four *P. pastoris* colonies integrating a single copy of pGAPGUT1-CFE. The ratios were calculated with the determined E values (Pfaffl method) and by assuming E to be 2 (ΔΔC_T_ method w/o c).

Colony Nr.	*ZEO* C_T_	*GUT1* C_T_	*ARG4* C_T_	Pfaffl Method		ΔΔC_T_ Method w/o c.	
				*ZEO:ARG4*	*GUT1:ARG4*	*ZEO:ARG4*	*GUT1:ARG4*
1	25.94 ± 0.16	26.03 ± 0.02	27.11 ± 0.08	-	-	-	-
2	25.31 ± 0.11	25.38 ± 0.08	26.54 ± 0.12	1.1	1.1	1.0	1.1
3	24.06 ± 0.09	24.42 ± 0.16	25.65 ± 0.49	1.4	1.3	1.3	1.1
4	25.21 ± 0.09	25.61 ± 0.11	26.11 ± 0.30	0.8	0.7	0.8	0.7

**Table 3 life-13-01885-t003:** Design proposal and experimental results of the three-level two-factor full factorial design. The relative activity (RA%) is the fumonisin esterase activity of the fermentation supernatants compared to the activity of the fermentation supernatant in YPD (1% yeast extract, 2% peptone, 4% dextrose) medium.

Run No.	Inactivated Yeast (%)	Dextrose (%)	Relative Activity (%)
1	6	6	95.3
2	3	6	101.2
3	6	2	73.9
4	6	4	85.0
5	3	4	93.7
6	0	4	17.0
7	3	4	97.2
8	0	2	17.7
9	0	6	14.4
10	3	4	97.7
11	3	2	81.8

**Table 4 life-13-01885-t004:** ANOVA for response surface quadratic model for CFE production using different compositions of the media. IY% = Inactivated yeast %; S% = dextrose %; L = linear effect; Q = quadratic effect; 1L by 2L = the interaction between the Inactivated yeast % linear and dextrose % linear effects; SS = sums of square; df = degree of freedom; MS = mean square; F = F-value; *p* = *p*-value, *p* < 0.05 is significant.

Factor	SS	df	MS	F	*p*
IY% (L)	7011.00	1	7011.00	1384.66	0.001
IY% (Q)	5234.50	1	5234.50	1033.81	0.001
S% (L)	234.38	1	234.38	46.29	0.021
1L by 2L	152.52	1	152.52	30.123	0.032
Lack of Fit	63.68	4	15.92	3.14	0.256
Pure Error	10.13	2	5.06		
Total SS	12,706.21	10			

**Table 5 life-13-01885-t005:** Regression coefficients of the quadratic model for CFE production. IY% = Inactivated yeast %; S% = dextrose %; L = linear effect; Q = quadratic effect; 1L by 2L = the interaction between the Inactivated yeast % linear and dextrose % linear effects; Regr. Coeff. = regression coefficient; Std. Err. = standard error; t(2) = t-probe; *p* = *p*-value, *p* < 0.05 is significant.

Factor	Regr. Coeff.	Std. Err.	t(2)	*p*
Mean/Intercept	16.22	3.18	5.1	0.036
IY% (L)	36.48	1.22	30.0	0.001
IY% (Q)	−4.87	0.15	−32.15	0.001
S% (L)	0.0375	0.73	0.05	0.960
1L by 2L	1.03	0.19	5.49	0.032

**Table 6 life-13-01885-t006:** Calculation of media costs based on prices in August 2023 in Hungary.

Run No.	Relative Activity (%)	Price vs. YPD (%)	Price vs. Centrum (%)
YPD	100.0	100	1761
1	95.3	7	139
2	101.2	6	115
3	73.9	6	109
4	85.0	6	124
Centrum (5, 7, 10)	96.2	5	100
6	17.0	4	76
8	17.7	3	61
9	14.4	5	91
11	81.8	4	85
Optimal	90.5	5	87

## Data Availability

Not applicable.

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
