# Peer review of "Optimization Workflow of Fumonisin Esterase Production for Biocatalytic Degradation of Fumonisin B1"

_life, 2023, doi:10.3390/life13091885_

Round 1
Reviewer 1 Report
This manuscript describes an interesting work aiming at developing a P. pastoris expression strain for fumonisin esterase production. The enzyme was used for the biocatalytic degradation of fumonisin B1. Optimization of the protein production medium was also achieved by the authors using design of experiments (DOE) approach. The work is interesting and within the scope of the journal. However, there are a few points which need to be addressed by the authors:
1) What was the expression level of the enzyme fumonisin esterase in the optimised medium. An SDS PAGE picture would be beneficial to the readers.
2) How the achieved expression of fumonisin esterase is compared with the common expression system using methanol oxidase promotor.
3) Did the author assess a downstream processing protocol for fractionation pf the enzyme ?
4) The conclusion section is long. The authors in this section need very briefly summarize and conclude the main findings. The other text can be transferred to the discussion section.
5) Line 499: delete underline
Author Response
This manuscript describes an interesting work aiming at developing a P. pastoris expression strain for fumonisin esterase production. The enzyme was used for the biocatalytic degradation of fumonisin B1. Optimization of the protein production medium was also achieved by the authors using design of experiments (DOE) approach. The work is interesting and within the scope of the journal. However, there are a few points which need to be addressed by the authors:
1) What was the expression level of the enzyme fumonisin esterase in the optimised medium. An SDS PAGE picture would be beneficial to the readers.
Response: We added the expression level of the optimum point of the DOE to the corresponding paragraph, to line 474 on page 11.: “This corresponds to 21.3 mg L-1 expression level, calculated by using the specific activity of CFE described in our parallel study [41].” We also added a picture of the SDS PAGE of the DOE in the Appendix Figure A2, and also references pointing the reader to it to lines 303 and 442.
2) How the achieved expression of fumonisin esterase is compared with the common expression system using methanol oxidase promotor.
Response: Question 2 of the reviewer is highly relevant to the subject of the work, however the pAOX1 production system is out of the scope of this work. For the same CFE protein, we described the expression of fumonisin esterase with the pAOX1 system in a separate work submitted in parallel [41]. The following has been added to the text between lines 509 and 512: “This expression level of CFE is comparable to the one obtained in our parallel study [41] using a methanol-induced pAOX1 expression system in an unoptimized one-week-long lab-scale fermentation (according to the protocol of ATUM [18]) in Buffered Complex Glycerol Medium (BMGY).”
3) Did the author assess a downstream processing protocol for fractionation pf the enzyme ?
Response: We thank the interest of the reviewer in the subject. However, the downstream processing of the fractionation of the enzyme was not assessed in this work. Our future communication on the results of crystallographic investigation of the CFE will include this information.
4) The conclusion section is long. The authors in this section need very briefly summarize and conclude the main findings. The other text can be transferred to the discussion section.
Response: We appreciate the feedback. Initially the conclusion section was 303 worlds long, which was shortened to 119 worlds in the revised version of the article.
5) Line 499: delete underline
Response: Thank you for your thoroughness, we corrected this editing error.

Reviewer 2 Report
Authors presented the fermentation optimization workflow for the production of fumonisin esterase. Authors clearly presented the procedures from stain engineering to the fermentation media optimization. After the workflow, the production yield of fumonisin esterase was significantly enhanced. Besides, authors further considered about the commerial cost for the fermentaion during the optimization workflow. Therefore, the workflow has the potential for industrial application.
One comment I have for the manuscript is about the activity assay of the esterase. The procedure is not quantitative. For instance, authors should indicate how much amount of culture (or diluted culture) were added so as to make comparison between different batch of fermentations. Did author consider the cell culture density during the activity assay. Without quantification, it is not convincing to compare the RA%.
Besides, authors presented SDS-PAGE method. However, there was no any gel pictures in the manuscript.
Authors should put at least one figure about the HPLC traces showing how the enzymatic assay was quantified.
Author Response
Authors presented the fermentation optimization workflow for the production of fumonisin esterase. Authors clearly presented the procedures from stain engineering to the fermentation media optimization. After the workflow, the production yield of fumonisin esterase was significantly enhanced. Besides, authors further considered about the commerial cost for the fermentaion during the optimization workflow. Therefore, the workflow has the potential for industrial application.
One comment I have for the manuscript is about the activity assay of the esterase. The procedure is not quantitative. For instance, authors should indicate how much amount of culture (or diluted culture) were added so as to make comparison between different batch of fermentations.
Response: We thank the reviewer for this insightful comment, we were not clear enough in our description of the materials and methods. Although section 2.9.2 starting from lane 304 describes the Enzyme Activity Measurement specifying in line 308 that “The enzymatic reaction was carried out in a plastic tube of 200 µL, using 72.2 µL of the FB1 solution, 7.8 µL of buffer A, and 20 µL of enzyme solution.” but the dilutions factors of the fermentation supernatants were not explicitly given in the submitted manuscript. To clarify this situation Table A3 has been added to the revised manuscript (line 311 contains the reference to this table: “Table A3 shows the dilution factors of each fermentation analyzed in this work.”). Addition of this information to manuscript completes the description of procedure to be fully reproducible, and allows a clear comparison between the batches.
Did author consider the cell culture density during the activity assay. Without quantification, it is not convincing to compare the RA%.
Response: Indeed, the cell culture density is a highly important factor influencing the activity assay. Unfortunately, due to the presence of the inactivated yeast in the media, it was impossible to accurately quantify the cell culture density with the traditional methods, such as OD measurement or wet cell mass. Nonetheless, qualitative observations during the experiments noted, that in the case of fermentations resulting in above RA% of 70%, the variability of the cell culture density was low, therefore the resulting error was negligible.
Besides, authors presented SDS-PAGE method. However, there was no any gel pictures in the manuscript.
Response: Thank you for your comment, we added the missing SDS-PAGE picture to the Appendix, as Figure A2, and also references pointing the reader to it in lines 303 and 442.
Authors should put at least one figure about the HPLC traces showing how the enzymatic assay was quantified.
Response: We appreciate your suggestion and added the HPLC trace to the Appendix Figure A3, and a reference to the HPLC trace in line 333 “A representative HPLC trace of the analysis of the enzymatic hydrolysis of FB1 is depicted in Figure A3.”.

Reviewer 3 Report
Manuscript contains valuable results of optimization of novel expression system for production of important industrial enzyme with respect to produced catalytic activity and production costs. Another quality of manuscript is that results scale up of enzyme production was performed. Manuscript is well organized and authors discussed results in adequate manner. Hence, my recommendation is to accept manuscript for publication.
Author Response
Manuscript contains valuable results of optimization of novel expression system for production of important industrial enzyme with respect to produced catalytic activity and production costs. Another quality of manuscript is that results scale up of enzyme production was performed. Manuscript is well organized and authors discussed results in adequate manner. Hence, my recommendation is to accept manuscript for publication.
Response: We appreciate your comments and review.

Round 2
Reviewer 1 Report
The revised version of the manuscript was improved and can be accepted for publication.